# Heat Transfer Through Insulating Glass Units Subjected to Climatic Loads

**DOI:** 10.3390/ma13020286

**Published:** 2020-01-08

**Authors:** Zbigniew Respondek

**Affiliations:** Faculty of Civil Engineering, Czestochowa University of Technology, 42-201 Częstochowa, Poland; zbigniew.respondek@pcz.pl

**Keywords:** glass in building, insulating glass units, heat loss in buildings, climatic loads

## Abstract

One of the structural elements used in the construction of insulating glass units (IGUs) are tight gaps filled with gas, the purpose of which is to improve the thermal properties of glazing in buildings. Natural changes in weather parameters: atmospheric pressure, temperature, and wind influence the gas pressure changes in the gaps and, consequently, the resultant loads and deflections of the component glass panes of a unit. In low temperature conditions and when the atmospheric pressure increases, the component glass panes may have a concave form of deflection, so that the thickness of the gaps in such loaded glazing may be less than its nominal thickness. The paper analyses the effect of reducing this thickness in winter conditions on the design heat loss through insulating glass units. For this purpose, deflections of glass in sample units were determined and on this basis the thickness of the gaps under operating conditions was estimated. Next, the thermal transmittance and density of heat-flow rate determined for gaps of nominal thickness and of thickness reduced under load were compared. It was shown that taking into account the influence of climatic loads may, under certain conditions, result in an increase in the calculated heat loss through IGUs. This happens when the gaps do not transfer heat by convection, i.e., in a linear range of changes in thermal transmittance. For example, for currently manufactured triple-glazed IGUs in conditions of “mild winter”, the calculated heat losses can increase to 5%, and for double-glazed IGUs with 10–14 mm gaps this ratio is about 4.6%. In other cases—e.g., large thickness of the gaps in a unit, large reduction in outside temperature—convention appears in the gaps. Then reducing the thickness of the gaps does not worsen the thermal insulation of the glazing. This effect should be taken into account when designing IGUs. It was also found that the wind load does not significantly affect the thickness of the gaps.

## 1. Introduction

Generally used in the building industry as a filling of windows or glass facades, insulating glass units (IGUs) consist of two or more component glass panes, connected at the edges with a glass spacer. The space between the component glass panes forms a tight gap filled with gas. In order to improve the thermal performance of the building partition constructed this way, the gap is filled with gas with lower thermal conductivity than air, most often argon. Further improvement of the performance is achieved by the use of component glass panes with a low-E coating—such a coating must be located on the side of the gap because it corrodes quickly when exposed to weather conditions. The tightness of the gap in insulating glass units is therefore a necessary factor to maintain good thermal insulation of transparent glazing [1,2].

Tight gaps, however, determine some specific properties of IGUs in the context of environmental loads transfer and the associated deformation of structural elements. The gap is filled with gas in the production process of the unit, therefore the gas in the gap has some initial parameters of pressure, temperature, and volume. Under operating conditions, an insulating glass unit is exposed to climatic loads which generate loads and deflections of the component glass panes due to the pressure difference between the gap and the environment. For example, an increase in atmospheric pressure or a decrease in the gas temperature in the gap results in a concave form of deflection of the panes (Figure 1a) and the opposite changes of these parameters in a convex form (Figure 1b). The magnitude of the under or overpressure in the gap depends not only on the value of climatic loads, but also on the structure of IGU. In general, it increases with reduced IGU dimensions (width × length), increased thickness of the gas gap, and increased thickness of the component glass plates. How the pressure difference affects the deflections in IGU will be presented later in this article.

In the case of wind exposure (Figure 1c), the tightness of the gap has a positive effect on the load distribution in an IGU. Due to changes in gas pressure in the gaps, the external load is partly transferred to the other panes of the unit.

The deflections of the glass described above result in deformation of the image viewed in the light reflected from the glass in windows or on glass facades (Figure 2). It is important that under conditions of low air temperatures, i.e., during the heating season, insulating glass units tend to take the concave form of deflection. The result is a reduction in the thickness of the gas space—especially in the central part of the glazing, where the component glass panes are closest to each other—which makes it possible to reduce the thermal insulation of the IGU.

The aim of the analysis carried out in the paper was to determine the effect of taking into account the reduction in thickness of gas-filled gaps in insulating glass units in winter conditions on the calculated heat losses through these partitions. The analysis was made for example for double- and triple-glazed IGUs. A detailed numerical quantification of this phenomenon was carried out for various IGU constructions.

In the literature, studies describing previous research in this area can be found: Barnier and Bourret [3] analyzed the effect of plate curvature in IGUs on the thermal transmittance (*U*_g_-value). The authors determined the *U*_g_-value for IGUs with variable gap thickness (limited by the surfaces of deflected panes), considering the average gap thickness in the loaded IGU as reliable. The authors stated that this assumption becomes reasonable when plate curvature is small and it is certainly acceptable in the conduction regime, where the convective movement is not significant. This article provides the results of sample calculations for double- and triple-glazed units under winter conditions. It was found that taking into account the plate curvature increases the calculated *U*_g_-value from 4.4% to 5.8%. Calculations were also made to account for changes in weather conditions (typical meteorological year) for Montreal and Toulouse. The results indicate that *U*_g_ may vary up to 5% above and 10% below the yearly average.

Hart et al. [4] analyzed the *U*_g_-value calculated from real deflections of double and triple-glazed units, measured in summer and winter at several locations in the USA. It was found that a 20 °C temperature difference reduces thermal performance by 4.6% for double-glazed IGUs and by 3.6% for triple-glazed IGUs.

Penkova et al. [5] presented examples of numerical analysis and experimental research regarding both parameters related to heat flow and climate loads. However, no detailed analysis of the change in thermal transmittance related to the deflections in the IGUs was carried out.

Thermal imaging photographs illustrating a decrease in thermal insulation in the central part of the glazing were published as well [6]. Examples are presented where the temperature in the central part of the glazing is 1–3 °C higher than the average on its surface (in images from the outside). An example of an IGU in which the component panes came into contact due to climatic loads is also presented.

## 2. Methodology for the Calculation of Static Quantities in IGUs

The methods of calculation of static quantities in double-glazed IGUs loaded with climatic factors are described in the literature. Mention may be made here of analytical models presented in papers [7,8,9,10] and numerical models allowing for consideration of the possibility of non-linear deflections of component glass [11,12]. The results of calculations presented in this paper were obtained using the author’s analytical model proposed in the article [13], which allows to calculate the load and deflection of component glass panes in units with any number of tight gaps.

The basis for the calculation of static quantities in IGUs is the assumption that the gas in the gaps meets the ideal gas equation:(1)$$\frac{p_{0}\cdot v_{0}}{T_{0}}=\frac{p_{\mathrm{op}}\cdot v_{\mathrm{op}}}{T_{\mathrm{op}}}=\mathrm{const},$$
where:*p*_0_, *T*_0_, *v*_0_—initial gap gas parameters: pressure [kPa], temperature [K], volume [m^3^], obtained in the production process,*p*_op_, *T*_op_, *v*_op_—operating parameters—analogously.

It is also assumed that the glass panes are simply supported at the edges and that the linear dependence of deflection *w* [m] of the component glass pane on its resultant surface load *q* [kN/m^2^] is assumed. The latter assumption is considered to be sufficiently accurate if the deflection is not greater than the thickness of the glass [14]. The deflection function of a simply supported single pane of the *a* [m] width and *b* [m] length, subjected to the *q* [kN/m^2^] load, placed centrally in the x-y coordinate system, can be recorded as [15]:(2)$$w\left(x,y\right)=\frac{4qa^{4}}{\pi^{5}D}\sum_{i=1,3,5\ldots }\frac{{(-1)}^{(i-1)/2}}{i^{5}}\cos\frac{i\pi x}{a}\cdot \left(1-\frac{\beta_{i}\cdot \mathrm{th}\beta_{i}+2}{2\cdot \mathrm{ch}\beta_{i}}\cdot \mathrm{ch}\frac{i\pi y}{a}+\frac{1}{2\cdot \mathrm{ch}\beta_{i}}\cdot \frac{i\pi y}{a}\cdot \mathrm{sh}\frac{i\pi y}{a}\right),$$
with
(3)$$\beta_{i}=\frac{i\pi b}{2a},$$

*D* [kNm] is the flexural rigidity of glass pane:(4)$$D=\frac{E\cdot d^{3}}{12\cdot \left(1-\mu^{2}\right)},$$
where:*d*—is the glass pane thickness [m],*E*—is the Young’s modulus of glass [kPa],*μ*—is the Poisson’s ratio [-].

Change in gap volume ∆*v* [m^3^] resulting from the deflection of one of the limiting glass pane may be determined by integration of Equation (2):(5)$$\Delta v=\int_{-b/2}^{b/2}\int_{-a/2}^{a/2}w(x,y)dxdy,$$
(6)$$\Delta v=\frac{4qa^{6}}{\pi^{7}D}\sum_{i=1,3,5\ldots }\frac{{\left(-1\right)}^{(i-1)/2}}{i^{7}}\cdot \frac{\sin\frac{i\cdot \pi }{2}}{{\left(\mathrm{ch}\beta_{i}\right)}^{2}}\cdot \left(4\cdot \beta_{i}+2\cdot \beta_{i}\cdot \mathrm{ch}\left(2\cdot \beta_{i}\right)-3\cdot \mathrm{sh}\left(2\cdot \beta_{i}\right)\right),$$

After the relevant calculations have been made:(7)$$\Delta v={\alpha_{\prime }}_{v}\cdot \frac{q\cdot a^{6}}{D}=\alpha_{v}\cdot q,$$
where:

*α*′_v_—is the dimensionless coefficient dependent on the *b*/*a* ratio (Table 1) [-],*α*_v_—is the proportionality factor, [m^5^/kN].

Any change in climatic conditions (atmospheric pressure, temperature, wind) results in a change in the gas pressure in the gaps which affects the resultant operating load of each of the component glass panes. For each gap of an IGU it is possible to formulate the equation of state:(8)$$p_{0}\cdot v_{0}\cdot T_{\mathrm{op}}=p_{\mathrm{op}}\cdot \left(v_{0}+\sum \Delta v\right)\cdot T_{0},$$
where:

Σ∆*v*—is the change in gap volume caused by deflection of both panes limiting it [m^3^].

As already mentioned in the article, double- and triple-glazed IGUs were analyzed. In the remaining part, the parameters of the individual component glass panes and gaps were marked with appropriate indices (Figure 3). It is also assumed that loads and deflections are positive if they face the interior, i.e., from left to right as in Figure 3.

Taking into account the adopted markings and conventions, Equation (8) for a double-glazed IGU can be presented in the form:(9)$$\frac{p_{0}\cdot v_{01}\cdot T_{\mathrm{op}1}}{T_{0}}=p_{\mathrm{op}1}\cdot \left[\left(p_{\mathrm{op}1}-c_{\mathrm{ex}}\right)\cdot \alpha_{v,\mathrm{ex}}+\left(p_{\mathrm{op}1}-c_{\mathrm{in}}\right)\cdot \alpha_{v,\mathrm{in}}\right],$$
with
(10)$$c_{\mathrm{ex}}=p_{a}+q_{z,\mathrm{ex}},\;c_{\mathrm{in}}=p_{a}-q_{z,\mathrm{in}},$$
where:p_a_—current atmospheric pressure [kPa],*q*_z,ex_, *q*_z,in_—load per area from outer factors, primarily wind [kN/m^2^], almost always *q*_z,in_ = 0.

After the relevant transitions have been made:(11)$$B\cdot p_{\mathrm{op}1}^{2}-A\cdot p_{\mathrm{op}1}-\frac{p_{0}\cdot v_{01}\cdot T_{\mathrm{op}1}}{T_{0}}=0,$$
with
(12)$$A=c_{\mathrm{ex}}\cdot \alpha_{v,\mathrm{ex}}+c_{\mathrm{in}}\cdot \alpha_{v,\mathrm{in}}-v_{01},$$
(13)$$B=\alpha_{v,\mathrm{ex}}+\alpha_{v,\mathrm{in}}.$$

Equation (11) has one solution giving non-negative results:(14)$$p_{\mathrm{op}1}=\frac{A}{2\cdot B}+\sqrt{{\left(\frac{A}{2\cdot B}\right)}^{2}+\frac{p_{0}\cdot v_{01}\cdot T_{\mathrm{op}1}}{B\cdot T_{0}}}.$$

In the case of a triple-glazed unit, a system of quadratic equations should be solved:(15)$$\{\begin{array}{c} p_{\mathrm{op}1}\cdot \left[v_{01}+\left(p_{\mathrm{op}1}-c_{\mathrm{ex}}\right)\cdot \alpha_{v,\mathrm{ex}}+\left(p_{\mathrm{op}1}-p_{\mathrm{op}2}\right)\cdot \alpha_{v,1-2}\right]-\frac{p_{0}\cdot v_{01}\cdot T_{\mathrm{op}1}}{T_{0}}=0\\ p_{\mathrm{op}2}\cdot \left[v_{02}+\left(p_{\mathrm{op}2}-p_{\mathrm{op}1}\right)\cdot \alpha_{v,1-2}+\left(p_{\mathrm{op}2}-c_{\mathrm{in}}\right)\cdot \alpha_{v,\mathrm{in}}\right]-\frac{p_{0}\cdot v_{02}\cdot T_{\mathrm{op}2}}{T_{0}}=0 \end{array}$$

This system has no analytical solution, but it can be solved numerically by iteration.

After calculating the operating pressure *p*_op_ for each of the gaps, the resultant loading *q* for each of the component glass panes can be determined:
for a double-glazed IGU
(16)$$q_{\mathrm{ex}}=c_{\mathrm{ex}}-p_{\mathrm{op}1},\;q_{\mathrm{in}}=p_{\mathrm{op}1}-c_{\mathrm{in}},$$for a triple-glazed IGU
(17)$$q_{\mathrm{ex}}=c_{\mathrm{ex}}-p_{\mathrm{op}1},\;q_{1–2}=p_{\mathrm{op}1}-p_{\mathrm{op}2},\;q_{\mathrm{in}}=p_{\mathrm{op}2}-c_{\mathrm{in}}$$

Deflection *w*_c_ [mm] in the center of the glass pane can be determined by the formula:(18)$$w_{c}={\alpha^{\prime }}_{w}\cdot \frac{q\cdot a^{4}}{D}\cdot 1000,$$
where:

*α*′_w_—is the dimensionless coefficient dependent on the *b*/*a* ratio (Table 1) [-].

However, the average deflection of the component glass panes *w*_m_ [mm] was determined from the formula:(19)$$w_{m}=\frac{\Delta v}{a\cdot b}\cdot 1000.$$

## 3. Materials and Methods

Thermal transmittance *U*_g_ [W/(m^2^·K)] of IGUs was calculated on the basis of the methodology described in standard [16], and heat losses were expressed by density of heat-flow rate *Φ* [W/m^2^] from the formula:(20)$$\Phi =U_{g}\cdot \left(t_{i}-t_{e}\right),$$
where:

*t*_i_, *t*_e_—are the internal and external air temperature [°C].

The heat flow through an insulating glass unit is complex—through conduction, convection, and radiation. The thermal resistance of gas-filled gaps *R*_s_ [(m^2^·K)/W] has the greatest influence on the *U*-value. For each gap:(21)$$R_{s}=\frac{1}{h_{g}+h_{r}},$$
with
(22)$$h_{g}=\frac{\lambda_{{}_{g}}\cdot Nu}{s},$$
(23)$$h_{r}=\frac{4\cdot \sigma \cdot T_{m}^{3}}{\frac{1}{\varepsilon_{\mathrm{sur}1}}+\frac{1}{\varepsilon_{s\mathrm{ur}2}}-1},$$
where: *h*_r_—is the thermal conductance by radiation [W/(m^2^·K)],*h*_g_—is the thermal conductance of gas (by conduction and convection) [W/(m^2^·K)].*λ*_g_—is the thermal conductivity of gas [W/(m·K)],*s*—is the gas gap thickness [m],*Nu*—is the Nusselt number [-],σ—is the Boltzmann constant 5.6693 × 10^−8^ W/(m^2^·K^4^)*T*_m_—is the average temperature of both surfaces delimiting the gap [K],*ε*_sur1_, *ε*_sur2_—are the emissivity of surfaces delimiting the gap [-].

It is particularly important whether convection occurs in the gaps. In the case of narrow gaps (*Nu* < 1) it is assumed that convection does not occur—thermal insulation of the gap increases linearly with its thickness. If a certain gap thickness limit (for *Nu* = 1) is exceeded, the effect of convection is taken into account. In this non-linear range (for *Nu* > 1) thermal insulation of the IGU does not improve. The value of this thickness limit depends on many factors (see also [17,18]), first of all on: the type of gas; the calculation was based on the use of argon,location in the structure; the calculations assume a horizontal position, in units situated horizontally or diagonally convection increases.increasing the temperature difference on the surfaces of the glass panes limiting the gap affects the increase in convection,convection also increases when the average gas temperature in the gap increases.

The thermal resistance of the gaps is primarily influenced by the use of low-emission glass. Glass without coating has a standard coefficient of emission of *ε* = 0.837. Application of low-emission coating reduces the emissivity of the plate surface, which results in a significant reduction of heat transfer by radiation. Currently, in Central and Northern Europe, IGUs are most often produced, in which each gap is adjacent to one coated surface and one without coating (Figure 3). This solution is most often used in units currently produced in Central and Northern Europe. The values *ε*_sur1_ = 0.837 and *ε*_sur2_ = 0.04 were used in the calculations.

Glass conducts heat well, therefore the thickness of the glass panes has no significant effect on the *U*_g_-value. Physical parameters of argon were adopted on the basis of the standard [17].

Of course, thermal insulation is also affected by thermal surface resistance at the external side (*R*_e_ [(m^2^·K)/W]) and at the internal side of a window (*R*_i_ [(m^2^·K)/W]). They depend primarily on the positioning of the window in the structure and the velocity of air (a short analysis on this subject is presented in Chapter 5). The calculations assumed *R*_i_ = 0.13 (m^2^·K)/W (vertical position) and *R*_e_ = 0.04 (m^2^·K)/W (for wind velocity *V* = 4 m/s). These are often accepted comparative values, also in the standard [16].

The calculations according to the adopted model require the use of numerical methods, because we encounter several interdependent values here. For example, the temperature values of gas and glass surfaces depend on the temperature distribution in the cross-section of the IGU. This distribution depends on the resulting thermal resistance values. The results of calculations were obtained by iteration after building the appropriate spreadsheet, assuming the steady state of heat transfer.

Figure 4 shows the effect of gap thickness *s* [mm] on the design *U*_g_-value for double- and triple-glazed IGUs, with glass thickness *d* = 4 mm, assuming *t*_i_ = 20 °C and in two variants of the outside air temperature *t*_e_ = 0 °C and *t*_e_ = −20 °C. The dashed line was used to determine the limits of gap thickness at which *Nu* = 1.

Figure 4 shows that at low temperatures *t*_e_ the limit thickness decreases. It can also be stated that in the case of triple-glazed IGUs, the difference in temperature in the gap is smaller, and the thickness of the boundary increases.

## 4. IGUs Under Pressure and Temperature Changes—Presentation and Discussion of Test Results

An analysis of the influence of climate loads on heat loss through IGUs under winter conditions was carried out for sample units with dimensions 0.7 × 1.4 m. Glass material parameters were adopted according to the standard [19]: *E* = 70 GPa, *μ* = 0.2.

It was also assumed that the following initial parameters were obtained in the argon-filled gaps during the production process *T*_0_ = 20 °C = 293.15 K, *p*_0_ = 100 kPa. In these conditions, the component glass panes are flat.

Two variants of the temperature drop load were used.

*Variant 1.* Reduced temperature conditions: *t*_i_ = 20 °C, *t*_e_ = −20 °C; the gas temperature in each gap was calculated for each case based on the temperature distribution in the particular IGU: for a double-glazed IGU *T*_op1_ = −2.37 to −2.25 °C, for triple-glazed IGU *T*_op1_ = −10.09 to −9.66 °C, *T*_op2_ = 7.60 to 7.97 °C.

*Variant 2*. Conditions for a “mild winter”: *t*_i_ = 20 °C, *t*_e_ = 0 °C; gas temperature: for a double-glazed IGU *T*_op1_ = 8.80 to 9.01 °C, for triple-glazed IGU *T*_op1_ = 4.97 to 5.08 °C, *T*_op2_ = 13.66 to 14.10 °C.

First, the effect of varying glass thickness on the gap width in the loaded set was investigated. IGUs with 16 and 12 mm gap thickness were analyzed in various combinations of 3, 4, and 6 mm thick panes. It was assumed that IGUs are only loaded with the temperature drop, as in variant 1, i.e., the current atmospheric pressure *p*_a_ = *p*_0_ = 100 kPa. The results of the calculations are presented in Table 2.

The resultant loading *q*_ex_ and *q*_in_ (absolute value) illustrates the underpressure in the gaps in relation to atmospheric pressure. The parameter *q*_1-2_ is the difference in operating pressure between the gaps in a triple-glazed IGU. From Equations (18) and (19) the extreme deflection (in the center of the pane) *w*_c_ and the average deflection *w*_m_ (the *w*_m_ values are given between parentheses) were calculated for each pane.

On the basis of these deflections, the minimum gap thickness in the center of the IGU *s*_c_ [mm] and the average gap thickness *s*_m_ [mm] were calculated.

On the basis of the calculations presented in Table 2, it was found that the calculated values of *s*_c_ and *s*_m_ for analyzed IGUs with gaps of the same nominal thickness do not differ much from each other. This is despite the fact that the deflection of component glass panes varies considerably. The effect of gas interactions in tight gaps can be seen here. Rigid panes are less susceptible to deflection, but the external load is less compensated for by the gas pressure in the gap. After changing the thickness of all the panes in a unit, the load changes, but the deflections are similar. Therefore, when one of the glass panes changes to a stiffer one, the absolute load values of component glass panes increase, although their algebraic sum for each IGU is equal to 0. After such conversion, the less rigid panes deflect more because they are exposed to a higher loading—for this reason, a loaded IGU has approximately constant volume of gaps, despite the change in thickness of the component panes.

To identify the extent of the phenomenon described above, another example was solved. Figure 5 shows the influence of IGU width (at a constant ratio *b*/*a* = 2) on the maximum deflection of component panes *w*_c_ in double-glazed units at 3, 4 i 6 mm thick panes and 16 mm thick gap. It was assumed that IGU is loaded only by a change in atmospheric pressure by ∆*p* = *p*_a_ − *p*_o_ = 3 kPa. This means that the current atmospheric pressure is *p*_a_ = 103 kPa. It can be added here that the results of calculations of static quantities are not very sensitive to the value of *p*_o_, and to a significant extent to ∆*p*. This means that if we assumed, for example, *p*_o_ = 950 kPa i *p*_a_ = 980 kPa, the results would be almost identical.

Figure 5 shows that that greater diversity of deflections for units of different component panes thickness occurs in the case of smaller IGU sizes. Then, however, the deflection values are smaller and it can be expected that changes in the gap thickness are also small.

In the context of the above, further analysis was carried out for IGUs with the same thickness of component glass panes *d* = 4 mm and different gap thicknesses *s* were assumed. Table 3 presents calculations of *w*_c_, *w*_m_, *s*_c_, and *s*_m_ values for units loaded with temperature change as in variant 1 and simultaneously operating loading with external atmospheric pressure increase of ∆*p* = 3 kPa. These are particularly unfavorable operating conditions in the context of reducing the thickness of the gaps. An analogous calculation was carried out for variant 2 (Table 4).

Based on the above data, Table 5 (for variant 1) and Table 6 (for variant 2) present the results of calculations of the thermal transmittance *U* and the density of heat-flow rate *Φ* [W/m^2^]:*U*_g_, *Φ*_g_—describe heat loss without taking into account the curvature of the panes, calculated for the nominal thickness of the gaps,*U*_c_, *Φ*_c_—describe possible local heat loss near the IGU center, i.e., where the distance between the panes is the smallest, calculated for the thickness of the gaps *s*_c_,*U*_m_, *Φ*_m_—describe the average heat loss through the IGU, calculated for the thickness of the gaps *s*_m_.

Finally, the percentage increase in the calculated quantities is also presented (∆*Φ*_c_*,* ∆*Φ*_m_*)* for units of nominal gap thickness.

The data presented in Table 5 and Table 6 and Figure 4 indicate that the reduction in the thickness of the gaps of insulating glass units due to their deflection under a drop in gas temperature and a rise in atmospheric pressure may result in an increase in design heat losses in relation to the calculations without taking into account the curvature of the panes. The increase in heat loss occurs in the linear range of the *U*_g_-value change, i.e., when the conditions inside the gap lead to *Nu* < 1. It is different when the *U*_g_-value changes in the non-linear range (*Nu* > 1). Heat losses do not increase. Then, the reduction of gap thickness can lead to a slight decrease in the calculated *Nu* value, which translates into a slight reduction in the calculated heat losses.

In this context, it is preferable to design IGUs such that it has *Nu* > 1 with a certain margin based on glazing deflections. However, this task should be approached with great caution, taking into account local climate conditions. It is necessary to check if the thickening of the gaps between the panes will not lead to excessive overpressure during the summer, due to the heating of gas in the gaps.

One more feature of the described phenomenon should be noted. In the linear range of changes in the *U*_g_-value, the indices ∆*Φ_c_* and ∆*Φ*_m_ almost do not depend on the thickness of the gaps. This is due to the fact, as additional calculations have shown, that the relationship between the thickness of IGU gaps and static quantities (resultant loading of component glass and their deflections) is also linear.

For many years, double-glazed IGUs dominated the market. Currently, due to the need to save energy, in Central and Northern Europe, triple-glazed IGU 4-16-4-16-4 is the most commonly produced and sold glazing for windows. Figure 6 presents an analysis illustrating the dependence of the percentage change in the calculated heat loss ∆*Φ*_m_ for these units on their width (at a constant ratio *b/a* = 2), under different external temperature conditions *t*_e_. Simultaneous pressure increase ∆*p* = 3 kPa was assumed. Other data was used as in previous examples.

It was found that the described effect is important for the currently sold glazing in “mild winter” conditions, i.e., when the outside temperatures fluctuate within between −5 °C and 5 °C. For IGU width above 0.7m, the ratio ∆*Φ*_m_ changes from 3.9% to 5.0%. These values are characteristic of the average temperature during the winter months in many places around the world.

## 5. Notes on IGUs Wind Load

Wind pressure or suction are also factors that cause deflection of the component glass panes in an IGU. As already mentioned, the wind velocity pressure acts directly only on the outer pane, but due to the change in the gas pressure in the gaps, the resultant load is distributed over all the panes of the unit. Table 7 shows the resultant loads and deflections in sample unit’s surface loaded with 0.3 kN/m^2^, which approximately corresponds [20] to a pressure of wind with velocity *V* of approx. 80 km/h (22.2 m/s).

Table 7 demonstrates that in the majority of units the deflections of component glass have similar values. Greater variations may occur when thicker panes are used, but the deflection values are small. It can therefore be concluded that the change in the thickness of the gaps due to wind load is small and has no noticeable effect on heat loss by IGUs.

Wind velocity has an indirect effect on heat loss. It is a factor influencing external thermal surface resistance on the outside, which translates into the *U*_g_-value. Graphic illustration of this effect is shown in Figure 7. The calculations were made for units with gap thickness of 16 mm. It can be noted that in the case of triple-glazed IGUs, the effect of wind velocity is negligible.

## 6. Conclusions

One of the factors influencing thermal transmittance *U*_g_ of insulating glass units is the thickness of gas-filled tight gaps. It is assumed in the calculation procedures that this thickness is not dependent on temporary changes in climatic factors. The thickness is variable under real operating conditions. In winter conditions in particular, IGU component glass panes take a concave form of deflection, which reduces the thickness of the gaps. This effect increases if the atmospheric pressure increases at the same time.

Based on the example calculations carried out, it has been shown that the increase in the calculated heat losses associated with the reduction of the gap thickness occurs when the conditions in the gap lead to *Nu* < 1, i.e., when the thermal transmittance of the gas layer is linearly dependent on its thickness. Heat losses can then increase to about 4.6% for double-glazed IGUs and to about 5% for triple-glazed ones, for external air temperature *t*_e_ = 0 °C. These values almost do not depend on the nominal thickness of the gaps, which results from the linear dependence of static quantities in an IGU on this thickness. Under certain conditions, heat losses calculated according to standard procedures may therefore be underestimated.

It is different in the non-linear range of the *U_g_*-value change (*Nu* > 1), i.e., when the outside temperature drops significantly or the gaps are thick enough. The thermal performance of glazing does not deteriorate. It is therefore advantageous to design IGUs so that *Nu* > 1, but it is necessary to take into account local climatic conditions and analyze loads that may also occur during the summer period.

In the case of the most commonly sold triple-glazed units 4-16-4-16-4 heat losses may be underestimated when the outside temperatures fluctuate between −5 °C and 5 °C. For large IGU dimensions, the ∆*Φ*_m_ index totals then from 3.9% to 5.0%.

It was also shown that the effect of wind load on gap thickness change is negligible in the context of heat loss estimation.

## Figures and Tables

**Figure 1 materials-13-00286-f001:**
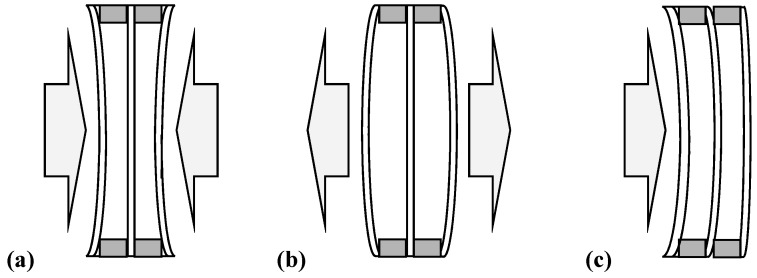
Typical deflections of insulating glass units: (**a**) concave form of deflection, (**b**) convex form of deflection, (**c**) deflection characteristic of wind load.

**Figure 2 materials-13-00286-f002:**
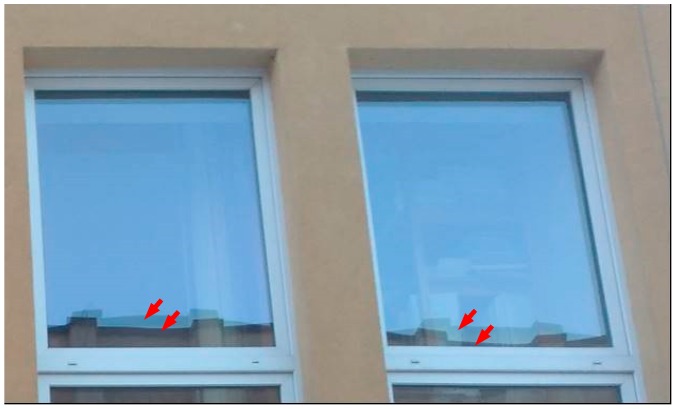
Visible distorted reflection of the image of the neighboring building from both insulating glass unit (IGU) component glass panes indicates the concave form of deflection of the unit.

**Figure 3 materials-13-00286-f003:**
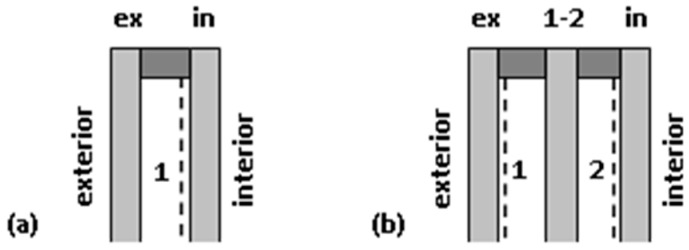
Index designations of IGU elements and location of low-emission coatings (dashed line): (**a**) double-glazed IGU, (**b**) triple-glazed IGU.

**Figure 4 materials-13-00286-f004:**
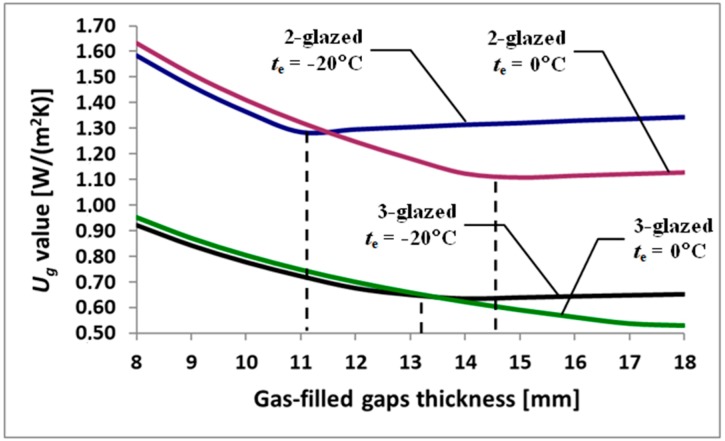
The dependence of the *U*_g_-values of IGUs on the thickness of the gaps.

**Figure 5 materials-13-00286-f005:**
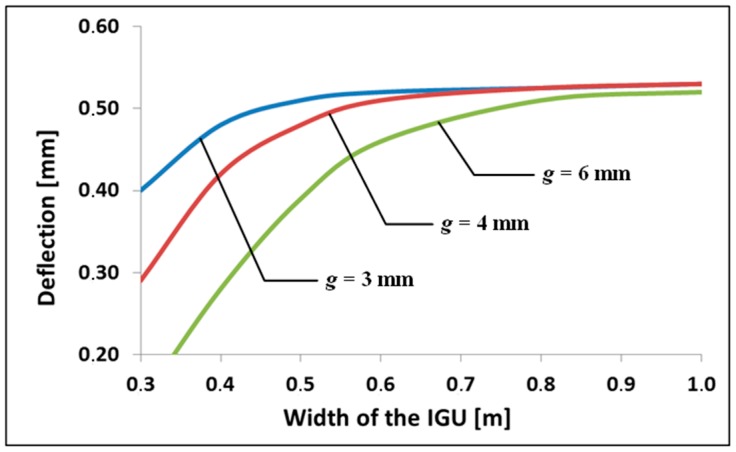
Dependence of the deflection of component glass panes *w*_c_ on the width of the IGU (atmospheric pressure increase of ∆*p* = 3 kPa).

**Figure 6 materials-13-00286-f006:**
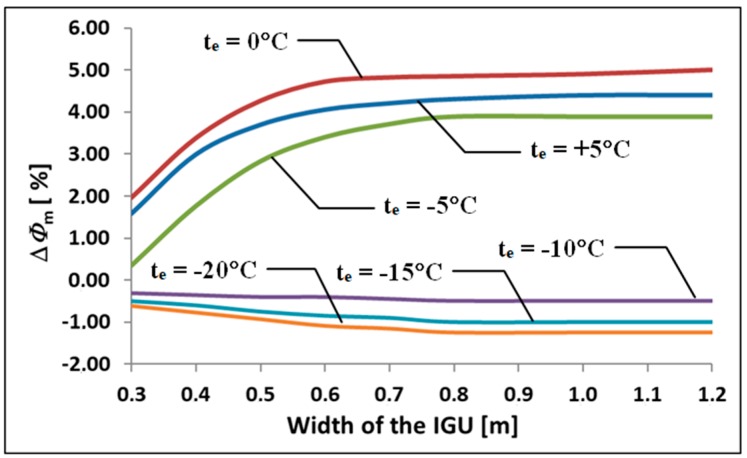
Dependence of the ∆*Φ*_m_ index on the width of the IGU and the outside temperature.

**Figure 7 materials-13-00286-f007:**
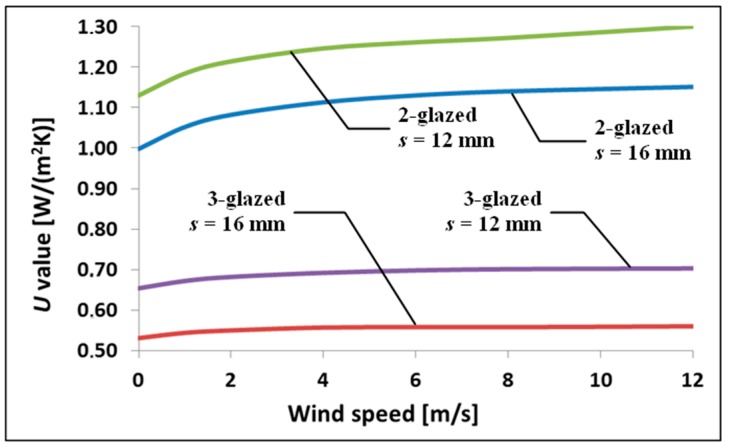
Influence of wind velocity on the *U*_g_-value of sample insulating glass units.

**Table 1 materials-13-00286-t001:** Coefficients for calculating volume change and deflection for simply supported glass pane.

***b/a***	1.0	1.1	1.2	1.3	1.4	1.5
***α*′_v_**	0.001703	0.002246	0.002848	0.003499	0.004189	0.004912
***α*′_w_**	0.004062	0.004869	0.005651	0.006392	0.007085	0.007724
***b/a***	1.6	1.7	1.8	1.9	2.0	3.0
***α*′_v_**	0.005659	0.006427	0.00721	0.008004	0.008808	0.017055
***α*′_w_**	0.008308	0.008838	0.009316	0.009745	0.010129	0.012233

**Table 2 materials-13-00286-t002:** Static quantities and gap thicknesses in IGUs—under reduced temperature conditions (Variant 1).

Structure of IGU [mm]	Resultant Loading *q* [kN/m^2^]			Deflection *w*_c_ (*w*_m_) [mm]			Resultant Thickness of Gap *s* [mm]			
	ex	1-2	in	ex	1-2	in	gap 1		gap 2	
							*s* _c1_	*s* _m1_	*s* _c2_	*s* _m2_
***d*_ex_-*s*_1_-*d*_in_**	**Double-glazed units**									
4-16-4	0.218	-	−0.218	1.36 (0.59)	-	−1.36 (−0.59)	13.28	14.82	-	-
6-16-4	0.330	-	−0.330	0.61 (0.27)	-	−2.08 (−0.90)	13.31	14.83	-	-
4-12-4	0.164	-	−0.164	1.03 (0.45)	-	−1.03 (−0.45)	9.94	11.10	-	-
6-12-4	0.251	-	−0.251	0.47 (0.20)	-	−1.57 (−0.68)	9.96	11.12	-	-
3-12-3	0.070	-	−0.070	1.04 (0.45)	-	−1.04 (−0.45)	9.92	11.10	-	-
***d*_ex_-*s*_1_-*d*_1-2_-*s*_2_-*d*_in_**	**Triple-glazed units**									
4-16-4-16-4	0.463	−0.117	−0.346	2.90 (1.26)	−0.73 (−0.32)	−2.16 (−0.94)	12.37	14.42	14.57	15.38
6-16-4-16-4	0.840	−0.311	−0.529	1.56 (0.68)	−1.94 (−0.85)	−3.34 (−1.44)	12.50	14.47	14.63	15.41
4-12-4-12-4	0.345	−0.087	−0.258	2.16 (0.99)	−0.54 (−0.24)	−1.61 (−0.70)	9.30	10.77	10.93	11.54
6-12-4-12-4	0.631	−0.233	−0.398	1.17 (0.51)	−1.46 (−0.63)	−2.49 (−1.08)	9.37	10.86	10.97	11.55
6-12-3-12-4	0.540	−0.112	−0.429	1.00 (0.43)	−1.66 (−0.72)	−2.68 (−1.17)	9.34	10.85	10.98	11.55
3-12-3-12-3	0.149	−0.037	−0.112	2.21 (0.96)	−0.55 (−0.24)	−1.66 (−0.72)	9.24	10.80	10.89	11.52

**Table 3 materials-13-00286-t003:** Static quantities and gap thicknesses in IGUs under reduced temperature conditions (Variant 1) and atmospheric pressure increase by ∆*p* = 3 kPa.

Structure of IGU [mm]	Resultant Loading *q* [kN/m^2^]			Deflection *w*_c_ (*w*_m_) [mm]			Resultant Thickness of Gap *s* [mm]			
	ex	1-2	in	ex	1-2	in	gap 1		gap 2	
							*s* _c1_	*s* _m1_	*s* _c2_	*s* _m2_
***d*_ex_-*s*_1_-*d*_in_**	**Double-glazed units**									
4-16-4	0.295	-	−0.295	1.84 (0.81)	-	−1.84 (−0.81)	12.32	14.38	-	-
4-14-4	0.259	-	−0.259	1.62 (0.70)	-	−1.62 (−0.70)	10.76	12.60	-	-
4-12-4	0.233	-	−0.233	1.39 (0.61)	-	−1.39 (−0.61)	9.22	10.78	-	-
4-10-4	0.187	-	−0.187	1.17 (0.51)	-	−1.17 (−0.51)	7.66	8.98	-	-
***d*_ex_-*s*_1_-*d*_1-2_-*s*_2_-*d*_in_**	**Triple-glazed units**									
4-16-4-16-4	0.613	−0.114	−0.500	3.83 (1.67)	−0.71 (−0.31)	−3.13 (−1.36)	11.46	14.02	13.58	14.95
4-14-4-14-4	0.540	−0.100	−0.440	3.38 (1.47)	−0.63 (−0.27)	−2.75 (−1.20)	9.99	12.26	11.88	13.07
4-12-4-12-4	0.460	−0.085	−0.375	2.87 (1.25)	−0.53 (−0.23)	−2.35 (−1.02)	8.60	10.52	10.18	11.21
4-10-4-10-4	0.387	−0.070	−0.317	2.42 (1.05)	−0.44 (−0.19)	−1.98 (−0.86)	7.14	8.76	8.46	9.33

**Table 4 materials-13-00286-t004:** Static quantities and gap thicknesses in IGUs under conditions for a “mild winter” (Variant 2) and atmospheric pressure increase by ∆*p* = 3 kPa.

Structure of IGU [mm]	Resultant Loading *q* [kN/m^2^]			Deflection *w*_c_ (*w*_m_) [mm]			Resultant Thickness of Gap *s* [mm]			
	ex	1-2	in	ex	1-2	in	gap 1		gap 2	
							*s* _c1_	*s* _m1_	*s* _c2_	*s* _m2_
***d*_ex_-*s*_1_-*d*_in_**	**Double-glazed units**									
4-16-4	0.188	-	−0.188	1.18 (0.51)	-	−1.18 (−0.51)	13.64	14.98	-	-
4-14-4	0.165	-	−0.165	1.03 (0.45)	-	−1.03 (−0.45)	11.94	13.10	-	-
4-12-4	0.143	-	−0.143	0.89 (0.40)	-	−0.89 (−0.40)	10.22	11.20	-	-
4-10-4	0.120	-	−0.120	0.75 (0.33)	-	−0.75 (−0.33)	8.50	9.34	-	-
***d*_ex_-*s*_1_-*d*_1-2_-*s*_2_-*d*_in_**	**Triple-glazed units**									
4-16-4-16-4	0.384	−0.058	−0.326	2.41 (1.04)	−0.36 (−0.16)	−2.04 (−0.89)	13.23	14.80	14.32	15.27
4-14-4-14-4	0.339	−0.050	−0.289	2.12 (0.92)	−0.31 (−0.14)	−1.81 (−0.79)	11.57	12.94	12.50	13.35
4-12-4-12-4	0.293	−0.042	−0.251	1.83 (0.80)	−0.26 (−0.11)	−1.56 (−0.68)	9.91	11.09	10.70	11.43
4-10-4-10-4	0.246	−0.035	−0.212	1.54 (0.67)	−0.22 (−0.09)	−1.32 (−0.57)	8.24	9.24	8.90	9.52

**Table 5 materials-13-00286-t005:** Quantities describing heat losses by IGUs under conditions of reduced temperature (Variant 1) and atmospheric pressure increase by ∆*p* = 3 kPa.

Gas Gap Thickness [mm]	Thermal Transmittance [W/(m^2^·K)]			Density of Heat-Flow Rate *Φ* [W/m^2^]			∆*Φ_c_* [%]	∆*Φ*_m_ [%]
	U	U_c_	U_m_	Φ	Φ_c_	Φ_m_		
**Double-glazed units**								
16	1.330	1.299	1.317	53.20	51.96	52.68	−2.3	−1.0
14	1.314	1.298	1.302	52.56	51.92	52.08	−1.2	−0.9
12	1.296	1.441	1.297	51.84	57.64	51.88	11.2	−0.1
10	1.364	1.630	1.467	54.56	65.20	58.68	19.5	7.6
**Triple-glazed units**								
16	0.643	0.653	0.636	25.72	26.12	25.44	1.6	−1.1
14	0.634	0.727	0.648	25.36	29.08	25.92	14.7	2.2
12	0.675	0.817	0.730	27.00	32.68	29.20	21.0	8.1
10	0.776	0.941	0.839	31.04	37.64	33.56	21.3	8.1

**Table 6 materials-13-00286-t006:** Quantities describing heat losses by IGUs under conditions for a “mild winter” (Variant 2) and atmospheric pressure increase by ∆*p* = 3 kPa.

Gas Gap Thickness [mm]	Thermal Transmittance [W/(m^2^·K)]			Density of Heat-Flow Rate *Φ* [W/m^2^]			∆*Φ_c_* [%]	∆*Φ*_m_ [%]
	U	U_c_	U_m_	Φ	Φ_c_	Φ_m_		
**Double-glazed units**								
16	1.113	1.142	1.107	22.26	22.84	22.14	2.6	−0.5
14	1.122	1.250	1.174	22.44	25.00	23.48	11.4	4.6
12	1.246	1.388	1.305	24.92	27.76	26.10	11.4	4.7
10	1.408	1.567	1.473	28.16	31.34	29.46	11.3	4.6
**Triple-glazed units**								
16	0.563	0.631	0.590	11.26	12.62	11.80	12.1	4.8
14	0.623	0.699	0.653	12.46	13.98	13.06	12.2	4.8
12	0.700	0.786	0.735	14.00	15.72	14.70	12.3	5.0
10	0.804	0.903	0.844	16.08	18.06	16.88	12.3	5.0

**Table 7 materials-13-00286-t007:** Static quantities in IGUs loaded with wind pressure of 0.3 kN/m^2.^

Structure of IGU [mm]	Resultant Loading *q* [kN/m^2^]			Deflection *w*_c_ [mm]		
	ex	1–2	in	ex	1–2	in
***d*_ex_-*s*_1_-*d*_in_**	**Double-glazed units**					
4-16-4	0.154	-	0.146	0.96	-	0.91
8-16-4	0.268	-	0.032	0.21	-	0.20
4-16-8	0.047	-	0.253	0.29	-	0.20
***d*_ex_-*s*_1_-*d*_1-2_-*s*_2_-*d*_in_**	**Triple-glazed units**					
4-16-4-16-4	0.109	0.098	0.092	0.68	0.61	0.58
8-14-4-14-4	0.247	0.028	0.026	0.19	0.17	0.16
4-12-4-12-8	0.053	0.039	0.208	0.33	0.24	0.16

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
