# Peer review of "Heat Transfer Through Insulating Glass Units Subjected to Climatic Loads"

_materials, 2020, doi:10.3390/ma13020286_

Round 1

Reviewer 1 Report

Title : “insulating glass units”: what does this term refer to? Isn’t it just a glass unit? When is a glass unit considered as insulating? Is there a threshold?

General:

The analysis of results should be clearer. It is sometimes blurry. is the calculation of heat transfer valid when the two panes are not parallel? A literature review should highlight if the energy calculation proposed here are valid (based on average cavity size, or minimum). The results should be put in perspective of the manufactured windows to conclude on the sensitivity of this parameter.

Abstract :

Given some numbers regarding the magnitude of the effect

Introduction:

Could you give the order of magnitude of the under or overpressure in the cavity? Literature review: give the quantification of this effect by the different authors

Chapter 3:

Use the proper terminology: U_g for glazing U-value mention the standard names, not only reference “The calculations assume the location of coatings as in Fig. 3.” -> non, it depends on the climate typology “As far as wind velocity V is concerned, 4 m/s is assumed as standard.” -> explain why is it used for “If a certain gap thickness limit is exceeded (for Nu = 1), the effect of convection is taken into account, resulting in a deterioration of the design U-value.” -> this is not true when looking at fig. 4, as you need to reach Nu >1 to get the maximum thermal insulation. It does not improve, but it is still the solution with the lowest U-value

Chapter 4:

There is no need to have assumed temperature, they should be calculated Variant 1 and 2: what were the manufactured temperatures? Table 2 : what are the values between parenthesis? Atmospheric pressure increase of 3 kPa -> could you give an example of what could cause such an increase? Table 4 and 6: the effect on the U-value is only visible when the condition inside the cavity lead to Nu <1, isn’t it? This should be mentioned (non-linear effect). So the glazing should be designed to have Nu>1 with a certain margin based on the glazing movements. “An increase in the density of the heat flow occurs when there is no conventional heat exchange in the gaps” -> what is the meaning of this sentence? The author should based his conclusion on the glazing typology sold actually. What is the average cavity size? A small study should emphasize if this effect is important for the glazing sold actually.

Conclusion:

“The analysis carried out for sample units showed that heat losses by IGUs, calculated taking 297 into account the phenomenon described above, may be greater than the losses calculated for units of 298 nominal gap thicknesses under conditions of lack of convection in the unit gaps - convection is lower 299 when the outside air temperature rises, the nominal thickness of the gaps decreases and in 300 multi-glazed IGUs.” -> this sentence should be rephrased “ncreasing the temperature differential and 308 increasing the nominal thickness of the gaps increases convection, but reducing the actual thickness 309 of the gaps in a loaded unit limits this convection.” : is convection increasing or decreasing? This should be rephrased

Author Response

Thank you very much for your valuable remarks and suggestions. The manuscript has been corrected. I have marked the changes in the text in yellow.

Title : “insulating glass units”: what does this term refer to? Isn’t it just a glass unit? When is a glass unit considered as insulating? Is there a threshold?

The term "insulating glass units (IGUs)" or "insulated glazing" is used as a generic name for sets of glass panes with a sealed gap. When these constructions came into use, the word "insulating" was to emphasize the better thermal parameters of this construction in relation to individual glazing. I also think that this wording is outdated, in my opinion a better term would be "complex glass panels". However, the terms "insulating glass units", “IGUs” are commonly used in literature and standards, so I suggest leaving this.

The analysis of results should be clearer. It is sometimes blurry.

I have reviewed the text in this context and reworded some paragraphs, especially in Chapter 4 and in the Conclusions. I have also corrected technical errors.

Is the calculation of heat transfer valid when the two panes are not parallel? A literature review should highlight if the energy calculation proposed here are valid (based on average cavity size, or minimum).

Calculations of the average thermal transmittance of IGUs based on the average size of the cavity was considered to be a sufficient approximation in the article by Barnier and Bourret [3]. I have supplemented it in the text (lines 78-81). Of course, I see here a field for more detailed research, especially in the non-linear range of changes in thermal transmittance.

In my article I have also calculated parameters describing possible local heat loss near the IGU center, i.e. where the distance between the panes is the smallest (lines 292-293). I think that assuming minimal gap thickness as valid is a sufficient approximation for the needs of this article, especially in the linear range of changes Ug.

The results should be put in perspective of the manufactured windows to conclude on the sensitivity of this parameter.

I have conducted a short analysis for the triple-glazed window glazing currently most commonly produced in Central and Northern Europe (lines 331-343).

Given some numbers regarding the magnitude of the effect.

This has been done in the Abstract (lines 22-24).

Could you give the order of magnitude of the under or overpressure in the cavity?

The pressure difference between the IGU gaps and the environment is very wide. For example, under load connected with a decrease in temperature and an increase in atmospheric pressure, as in the article, IGU 4-16-4 with size of 0.3´0.6 m has the underpressure of 4.6 kPa, while one with dimensions of 1.2´2.4 m has 0.03 kPa. It depends not only on the value of climatic loads - it can be generally stated that small-size IGUs, a units with thick panes and a units with thick gas gaps have greater over or underpressure. Of course, this translates into static quantities (deflection, stress) in the component panes in different ways. I did not analyze this in detail in this article, because this is not its main topic. I put the sentence in the Introduction (lines 49-53), and references to these relationships appear in the article.

In my calculations, the underpressure is illustrated by the parameter "resultant loading q" shown in Tables 2, 3 and 4. I supplemented the text with an appropriate explanation (lines 246-248).

Literature review: give the quantification of this effect by the different authors.

I have supplemented the literature review with this information (lines 76-98).

Use the proper terminology: U_g for glazing U-value mention the standard names, not only reference.

I have corrected it in the text.

The calculations assume the location of coatings as in Fig. 3.” -> non, it depends on the climate typology.

I have rewritten this fragment (lines 200-203).

As far as wind velocity V is concerned, 4 m/s is assumed as standard.” -> explain why is it used for.

I have explained it in the text (lines 206-211).

“If a certain gap thickness limit is exceeded (for Nu = 1), the effect of convection is taken into account, resulting in a deterioration of the design U-value.” -> this is not true when looking at fig. 4, as you need to reach Nu >1 to get the maximum thermal insulation. It does not improve, but it is still the solution with the lowest U-value.

I have rewritten this sentence (lines 187-189).

There is no need to have assumed temperature, they should be calculated.

I have recalculated everything again using Top1 i Top2 temperatures resulting from the temperature distribution in the glazing, not the assumed values (Tables 2÷6).

Variant 1 and 2: what were the manufactured temperatures? Table 2 : what are the values between parenthesis?

I have described it in the text (lines 233 and 249).

Atmospheric pressure increase of 3 kPa -> could you give an example of what could cause such an increase?

The increase in atmospheric pressure may be due to natural changes in weather conditions. I have explained it in the text (lines 269-274).

Another reason for the change in pressure may be the change in altitude. The atmospheric pressure values given in weather forecasts are related to sea level. Inland, the pressure is lower, approximately 1.2 kPa for every 100 m height difference. So, if the factory is located at a different height than the place of IGU installation, the unit may be exposed to high long-term climate loads. This problem has been noticed - now IGUs can be equipped with devices (valves) enabling pressure equalization in IGUs after transporting them to the place of installation.

Table 4 and 6: the effect on the U-value is only visible when the condition inside the cavity lead to Nu <1, isn’t it? This should be mentioned (non-linear effect). So the glazing should be designed to have Nu>1 with a certain margin based on the glazing movements.

This is true, I have included it in the text (lines 316-325 and 372-384).

“An increase in the density of the heat flow occurs when there is no conventional heat exchange in the gaps” -> what is the meaning of this sentence?

This sentence was unfortunately worded, I have deleted it.

The author should based his conclusion on the glazing typology sold actually. What is the average cavity size? A small study should emphasize if this effect is important for the glazing sold actually.

I have supplemented the text with a more detailed analysis of triple-glazed IGUs with a gap thickness of 16 mm. These are currently the most-sold window glazing in Central and Northern Europe (lines 331-343 and 385-387).

Sentences in Conclusions.

The Conclusions were reworded.

Reviewer 2 Report

The submitted manuscript discusses an interesting phenomenon related to double or triple glazed windows. The changes of thermodynamic states of the gas inside the closed cavities of the windows may cause pressure variances and consequently the variances of cavity thickness. This causes changes in thermal resistance of the gas cavity and, thus, heat losses of the windows may correspondingly increase or decrease.

This phenomenon is not new. Several other authors have investigated these phenomena in the past. The valuable feature of the submitted manuscript consists in the detailed numerical quantification of the phenomenon.

Conceptual remarks:

1) The part concerning the determination of pressure changes would deserve a detailed explanation, especially derivation of quadratic pressure equations. Reference [13], to which the author refers to,  does not contain such an explanation.

2) Equation (16) is incorrect (!). Instead of the wrong formula Φ = U/ (tite) should be the correct formula U=Φ/(ti – te). At first sight, this could invoke a feeling that many computations in the manuscript are wrong but, however, as can be checked in Tables 4 and 6, the computations have been performed according to the correct formula. Therefore, Equation (16) is a misprint and must be corrected.

3) At many places of the manuscript the U-value (transmittance) is mentioned and computed as if it were caused solely by the phenomena related to gas cavity. In fact, the U-values of windows are related not only to their gas cavities but they are also influenced, besides the conductivity of glass, by the conductances (i.e. coefficients of heat transfers) at the interior side (hi) and at the exterior side (he) of windows. This should be mentioned at some suitable place of the manuscript.

---------------------

Formal remarks:

1) Eq. (18) - λg not λs – see line 154.

2) Lines 168 – 169 : hardly understandable - reformulation needed.

3) Line 170 : hardly understandable - reformulation needed.

4) Line 211 : explain meaning for sc and sm – thickness in the center and average thickness. Symbols must be explained at the first place of their occurrence.

Author Response

Thank you very much for your valuable remarks and suggestions. The manuscript has been corrected. I have marked the changes in the text in yellow.

Conceptual remarks:

1) The part concerning the determination of pressure changes would deserve a detailed explanation, especially derivation of quadratic pressure equations. Reference [13], to which the author refers to,  does not contain such an explanation.

I have supplemented this in the text (lines 143-157)

2) Equation (16) is incorrect (!). Instead of the wrong formula Φ = U/ (ti – te) should be the correct formula U=Φ/(ti – te). At first sight, this could invoke a feeling that many computations in the manuscript are wrong but, however, as can be checked in Tables 4 and 6, the computations have been performed according to the correct formula. Therefore, Equation (16) is a misprint and must be corrected.

It is true, I have corrected it (line 169).

3) At many places of the manuscript the U-value (transmittance) is mentioned and computed as if it were caused solely by the phenomena related to gas cavity. In fact, the U-values of windows are related not only to their gas cavities but they are also influenced, besides the conductivity of glass, by the conductances (i.e. coefficients of heat transfers) at the interior side (hi) and at the exterior side (he) of windows. This should be mentioned at some suitable place of the manuscript.

I have included it in the text (lines 206-211 and 358-364).

Formal remarks:

1) Eq. (18) - λg not λs – see line 154.

I have corrected it (line 175).

2) Lines 168 – 169 : hardly understandable - reformulation needed.

I have corrected it (lines 194-195).

3) Line 170 : hardly understandable - reformulation needed.

I have corrected it (line 196).

4) Line 211 : explain meaning for sc and sm – thickness in the center and average thickness. Symbols must be explained at the first place of their occurrence.

I have explained it (lines 251-252).